# A library of polytypic copper-based quaternary sulfide nanocrystals enables efficient solar-to-hydrogen conversion

Liang Wu[1,4], Qian Wang[2,4], Tao-Tao Zhuang[1], Guo-Zhen Zhang [2], Yi Li[1], Hui-Hui Li[1], Feng-Jia Fan [3] & Shu-Hong Yu [1]✉

Designing polytypic homojunction is an efficient way to regulate photo-generated electrons and holes, thereafter bringing desired physical and chemical properties and being attractive photocatalysts for solar-to-hydrogen conversion. However, the high-yield and controllable synthesis of well-defined polytypes especially for multinary chalcogenide - the fundamental factor favoring highly efficient solar-to-hydrogen conversion - has yet to be achieved. Here, we report a general colloidal method to construct a library of polytypic copper-based quaternary sulfide nanocrystals, including $Cu_2ZnSnS_4$, $Cu_2CdSnS_4$, $Cu_2CoSnS_4$, $Cu_2MnSnS_4$, $Cu_2FeSnS_4$, $Cu_3InSnS_5$ and $Cu_3GaSnS_5$, which can be synthesized by selective epitaxial growth of kesterite phase on wurtzite structure. Besides, this colloidal method allows the precise controlling of the homojunction number corresponding to the photocatalytic performance. The single-homojunction and double-homojunction polytypic $Cu_2ZnSnS_4$ nanocrystal photocatalysts show 2.8-fold and 3.9-fold improvement in photocatalytic hydrogen evolution rates relative to the kesterite nanocrystals, respectively. This homojunction existed in the polytypic structure opens another way to engineer photocatalysts.

Semiconductor nanomaterials that can effectively serve as photocatalysts for solar-to-hydrogen production and degradation of non-biodegradable dyes have been drawn enough interests over the past decades[1–5]. In developing a special nanostructured photocatalyst, it is important to choose a semiconductor material that has a wide absorption spectral range and is environmentally sustainable and low-cost. Copper-based quaternary sulfide (CQS) semiconductors, which consist of earth-abundance elemental components and possess suitable bandgaps favoring big chance to get broad solar irradiation and

high light absorbance coefficients, are the most promising photocatalytic materials[6–17].

However, the fast recombination rate of photogenerated electrons and holes in a single CQS nanocrystal impedes solar-derived photocatalytic hydrogen evolution. Constructing heterojunction with type-II bandgap alignment is an efficient way to promote the photogenerated carrier separation in semiconductors[18–20], while the heterojunction with poor lattice match usually introduces defects at the interface which further traps charge carriers. Inspired by the homo-

[1]Department of Chemistry, Institute of Biomimetic Materials and Chemistry, Anhui Engineering Laboratory of Biomimetic Materials, Division of Nanomaterials and Chemistry, Hefei National Research Center for Physical Sciences at the Microscale, Institute of Energy, Hefei Comprehensive National Science Center, University of Science and Technology of China, Hefei 230026, China. [2]Department of Chemical Physics, iChEM (Collaborative Innovation Center of Chemistry for Energy Materials), Synergetic Innovation Center of Quantum Information and Quantum Physics, University of Science and Technology of China, Hefei 230026 Anhui, China. [3]CAS Key Laboratory of Microscale Magnetic Resonance and Department of Modern Physics, Synergetic Innovation Center of Quantum Information and Quantum Physics, University of Science and Technology of China, Hefei 230026 Anhui, China. [4]These authors contributed equally: Liang Wu, Qian Wang. ✉e-mail: shyu@ustc.edu.cn

junction in twined nanorods which exhibit enhanced photocatalytic performances[21–24], the homojunction in polytypes consists of chemically identical but structurally different materials, which match well at their interfaces, and can avoid the common problems associated with compositional changes and strain control at the interfaces, realizing efficient photogenerated carrier separation[25–29].

Recently, polytypes have been widely built in IV, III−V and II−VI semiconductors via vapor-liquid-solid growth and hydrothermal method[28,30–32]. Whereas, these methods are not suitable for synthesizing multinary chalcogenide nanocrystals. Colloidal synthesis method provides access to produce well-defined copper-based quaternary chalcogenide nanocrystals. Up to now, several polytypic copper-based ternary and quaternary chalcogenide nanocrystals have been prepared via colloidal method[33–38]. However, a general and facile method for the synthesis of polytypic CQS nanocrystals with exactly controlling the homojunction number has not been demonstrated so far, and the homojunction related photocatalytic performances need to be further explored.

Here, we report a general colloidal method which realizes the epitaxial growth of kesterite (KS) structure on the wurtzite (WZ) structure toward a library of polytypic CQS nanocrystals, including $Cu_2ZnSnS_4$ (CZTS), $Cu_2CdSnS_4$ (CCdTS), $Cu_2CoSnS_4$ (CCoTS), $Cu_2MnSnS_4$ (CMnTS), $Cu_2FeSnS_4$ (CFeTS), $Cu_3InSnS_5$ (CInTS) and $Cu_3GaSnS_5$ (CGaTS). Taking polytypic CZTS nanocrystals as an example, we can exactly control the homojunction number to prepare bullet-shaped single-homojunction polytypes (SHP, WZ-KS) and rugby-shaped double-homojunction polytypes (DHP, KS-WZ-KS). Moreover, the photocatalytic hydrogen evolution performances of polytypic CZTS nanocrystals were further examined. As a result,

polytypic CZTS nanocrystals exhibit higher photocatalytic activities than that of phase-pure CZTS nanocrystals with the same composition (270 μmol h⁻¹ g⁻¹ for SHP, 381 μmol h⁻¹ g⁻¹ for DHP, 145 μmol h⁻¹ g⁻¹ for WZ, 98 μmol h⁻¹ g⁻¹ for KS). Density functional theory (DFT) calculation studies further reveal that the polytypic homojunction has a type-II bandgap alignment, which is benefit for photogenerated carrier separation, resulting enhanced photocatalytic performance. The solar-to-hydrogen performances of other prepared polytypic CQS nanocrystals are further examined, and thus demonstrating our concept that polytypic homojunction can enhance the photocatalytic hydrogen evolution performances of semiconductors.

## Results

### Colloidal synthesis and characterization of polytypic CZTS nanocrystals

CZTS derived from ZnS, which inherits the atomic arrangement, has WZ and KS phases (Supplementary Fig. 1)[39,40]. The polytypic structures for CZTS are constructed with hexagonal stacking WZ in the [0001] direction and tetragonal stacking KS in the [112] direction. Thus, polytypic CZTS nanocrystals can be synthesized through colloidal epitaxial growth of KS phase on the {0001} facets of WZ structure along $[112]_{KS}/[0001]_{WZ}$ direction (Supplementary Figs. 2–4). Single-homojunction and double-homojunction polytypic CZTS nanocrystals are synthesized through selective epitaxial growth of KS phase on one (0001) facet or both two {0001} facets of WZ CZTS at appropriate reaction conditions, respectively (Supplementary Figs. 5–14).

Figure 1a, b show the morpholoy of the synthsized single-homojunction polytypic CZTS nanocrystals characterized by high-angle annular dark field scanning transmission electron microscopy

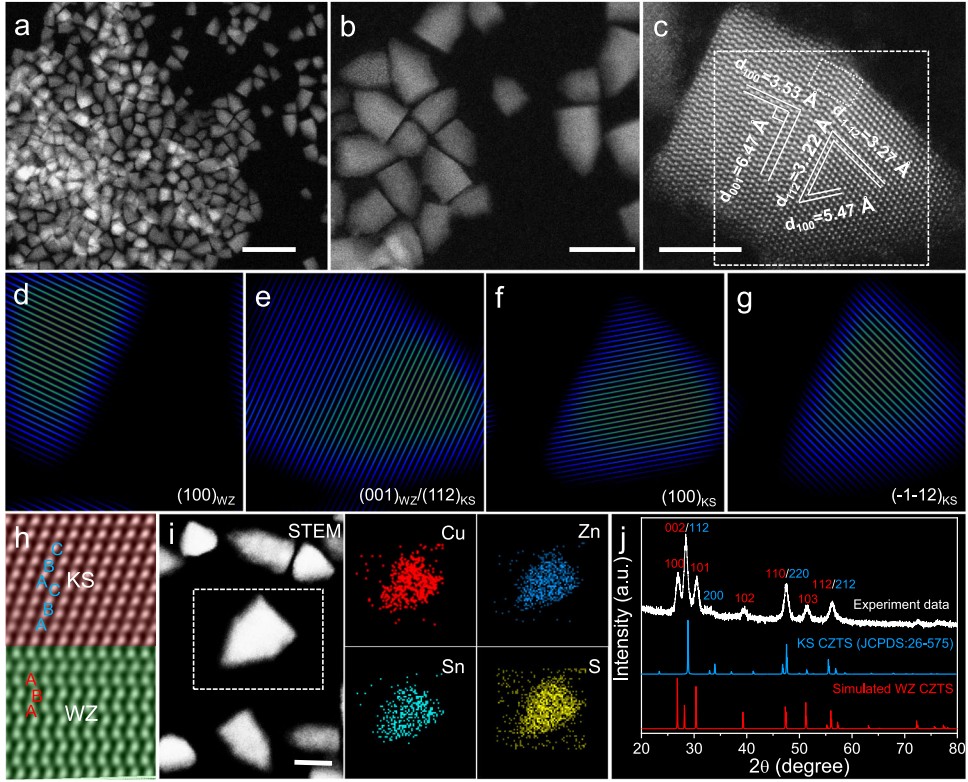

**Fig. 1 | Characterization of the single-homojunction polytypic CZTS nanocrystals.** Low-magnification HAADF-STEM (**a**) and high-magnification HAADF-STEM (**b**) images. **c** A representative aberration-corrected high-resolution HAADF-STEM image. Bragg-filtered images are derived using the $(100)_{WZ}$ (**d**), $(001)_{WZ}$/$(112)_{KS}$ (**e**), $(100)_{KS}$ (**f**) and $(-1-12)_{KS}$ (**g**) reflections by inverse FFT in selected area in **c**, respectively. **h** Enlarged high-resolution HAADF-STEM image in the selected area in **c**. **i** EDS element mapping of one randomly selected single-homojunction polytypic CZTS nanocrystal. **j** XRD pattern. For reference, the KS and simulated WZ XRD patterns of CZTS are showed below. Scale bars are 50 nm for **a**, 20 nm for **b** and **i**, and 5 nm for **c**, respectively.

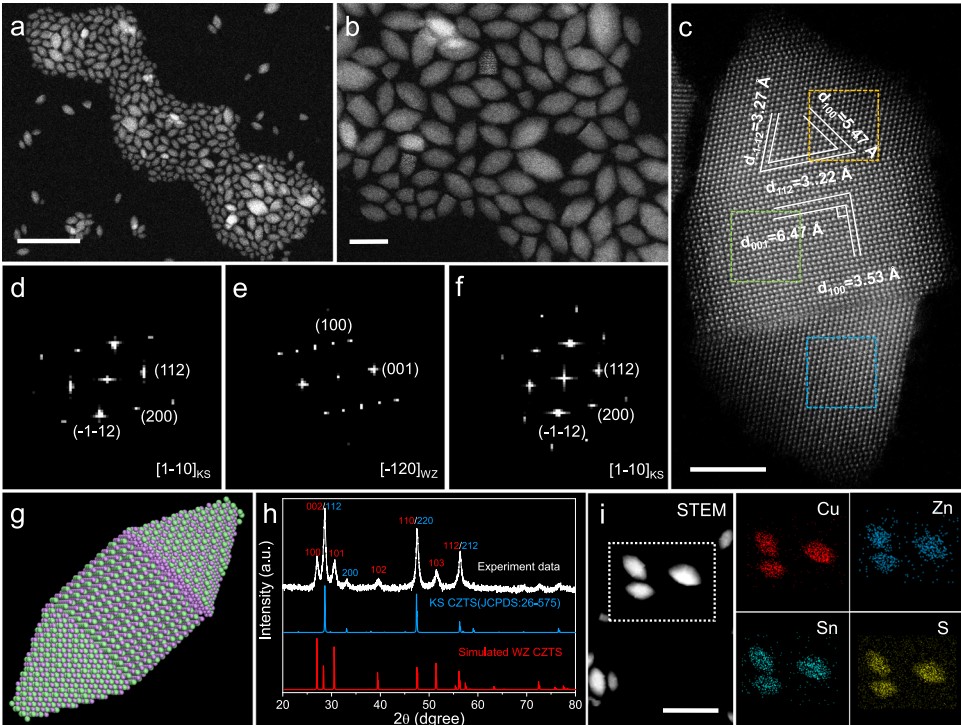

**Fig. 2 | Characterization of the double-homojunction polytypic CZTS nanocrystals.** Low-magnification HAADF-STEM (**a**) and high-magnification HAADF-STEM (**b**) images. **c** A representative aberration-corrected high-resolution HAADF-STEM image. FFT patterns of the corresponding orange (**d**), green (**e**) and blue (**f**) areas in **c**, respectively. **g** Schematic graph of a DHP nanocrystal. **h** XRD pattern. **i** EDS element mapping of three randomly selected double-homojunction polytypic CZTS nanocrystals. Scale bars are 50 nm for **a** and **i**, 20 nm for **b**, and 5 nm for **c**, respectively.

(HAADF-STEM), demonstrating that all the nanocrytals display a bullet-like shape. The representative aberration-corrected high-resolution HAADF-STEM was used to investigate the atomic stackings of a free-standing and individual single-homojunction polytypic CZTS nanocrystal. As shown in Fig. 1c, the interplane distances between the lattice fringes in the rectangle area are 6.47 and 3.53 Å, corresponding to the (001) and (100) planes of WZ CZTS nanocrystals, respectively. Furthermore, the lattice fringes of 3.22, 3.27 and 5.47 Å in the cusp are in consistence with the (112), (-1-12) and (100) planes of KS CZTS nanocrystals, respectively. Figure 1d, e shows the Bragg-filtered images from the (100) and (001) reflections of WZ CZTS, respectively. Figure 1e–g shows Bragg-filtered images from the (112), (100) and (-1-12) reflections of KS CZTS, respectively. The aforementioned results prove that the polytypic CZTS nanocrystals consist of WZ and KS structures. The enlarged high-resolution HAADF-STEM image in Fig. 1h clearly presents two different atom stacking forms separated by one homojunction, such as the A-B-A WZ atom stacking in the rectangle area and the A-B-C KS atom stacking in the cusp. Energy dispersive spectrometer (EDS) element mapping was used to characterize the element distribution in one randomly selected single-homojunction polytypic CZTS nanocrystals. Figure 1i shows that Cu, Zn, Sn and S elements homogeneously distribute in the polytypic nanocrystal, revealing that the homojunction is not due to the element heterogeneous distribution. The powder X-ray diffraction (PXRD) pattern (Fig. 1j) of the single-homojunction polytypic CZTS nanocrystals exhibits a typical WZ ZnS like diffraction pattern with enhanced (002), (110) and (112) peaks and an individual $(200)_{KS}$ peak, demonstrating the existence of WZ and KS structures and indirectly proving the polytypic structure of the obtained CZTS nanocrystals[41,42].

In addtion, the CZTS derives from ZnS through substitution of Zn atom with Cu and Sn atoms. We further demonstrated the influenc of zinc content on the polytypic structure of single-homojunction CZTS nanocrystals and found that—with controlled dosage of $Zn(AC)_2$—the

ratio of WZ phase can be modulated controllably. The WZ part in the obtained single-homojunction polytypic CZTS nanocrystals increased with zinc content until to a pure WZ structure with no exist of ZnS (Supplementary Figs. 15−20 and Supplementary Table 1). Interestingly, polytypic $Cu_2SnS_3$ nanocrystals could be obtained (Supplementary Figs. 15−17) without $Zn(AC)_2$. Besides, more than 3 g of single-homojunction CZTS nanocrystals could be synthesized after one single reaction (Supplementary Fig. 21).

Decreasing the dosage of 1-DDT to 0.5 ml, double-homojunction polytypic CZTS nanocrystals would be produced. The HAADF-STEM images in Fig. 2a, b show that the synthesized double-homojunction polytypic CZTS nanocrystals have a high purity of rugby shape. The representative aberration-corrected high-resolution HAADF-STEM image in Fig. 2c presents the detail atomic stackings of one randomly selected double-homojunction polytypic CZTS nanocrystal. The rugby-shaped polytypic nanocrystal consists of two KS cusps and one WZ part separated by two homojunctions. The interplane distances between lattice fringes shown in the rectangle area and one cusp are indexed to (001) and (100) planes of WZ and (112), (-1-12) and (100) planes of KS CZTS nanocrystals, respectively. These fringes agree with those in singe-homojunction polytypic CZTS nanocrystals. Selected-area fast Fourier transform (FFT) patterns (Fig. 2d−f) from the two ends and middle of the rugby-shaped nanocrystal match well with the typical $[1-10]_{KS}$-zone axis diffraction pattern of KS phase and the typical $[-120]_{WZ}$-zone axis diffraction pattern of WZ phase, respectively, certifying the polytypic structure of the obtained CZTS nanocrystals. Based on the said results, an atomic model of the KS-WZ-KS double-homojunction nanocrystal is schematically illustrated in Fig. 2g. PXRD was used to further detect the structure of the double-homojunction polytypic CZTS nanocrystals. As shown in Fig. 2h, a typical WZ structure PXRD pattern with enhanced (002), (110) and (112) peaks and an individual $(200)_{KS}$ peak clarifies the existence of two phases, indirectly proving the polytypic nanostructure of the obtained

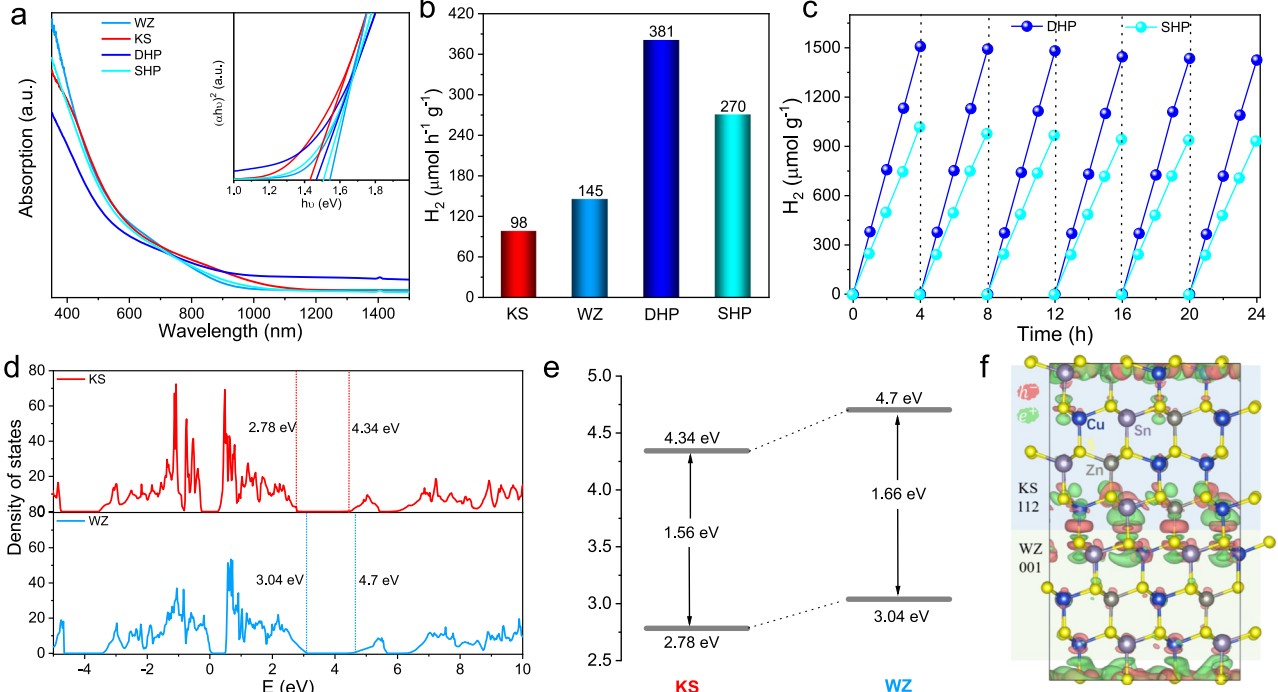

**Fig. 3 | Optical and photocatalytic properties of CZTS nanocrystals. a** UV-Vis-IR absorption spectra of the obtained polytypic and phase-pure CZTS nanocrystals. the insert shows the graph of linear extrapolation of $(\alpha h\upsilon)^2$ versus photon energy $(h\upsilon)$. **b** Photocatalytic hydrogen evolution rates of the obtained polytypic and phase-pure CZTS nanocrystals at room temperature under visible light $(\lambda > 420\ nm)$. **c** Cycles running of the polytypic CZTS nanocrystals for photocatalytic hydrogen evolution. After every 4 h, the produced hydrogen is evacuated. **d** The calculation Density of States of WZ and KS phases in the polytypic structures. **e** The bandgap alignments of polytypic nanocrystals. **f** The simulated charge distributions of the new hybrid combination.

CZTS nanocrystals. EDS element mapping of three randomly selected double-homojunction polytypic CZTS nanocrystals (Fig. 2i) illustrates that the homojunction does not owe to the heterogenous distribution of Cu, Zn, Sn, and S elements. The Raman spectra of synthesized polytypic CZTS nanocrystals demonstrate that there are no binary and ternary byproducts (Supplementary Fig. 22), indirectly proving the existence of homojunctions.

### Optical properties and photocatalytic performances of polytypic CZTS nanocrystals

The light harvesting ability is necessary for photocatalytic performance. To investigate the harvesting capability of sunlight, the absorption spectra of the synthesized polytypic and phase-pure CZTS nanocrystals are characterized by ultraviolet-vis-near infrared (UV-Vis-IR) absorption spectroscopy. Figure 3a depicts the UV-Vis-IR spectra of the single-homojunction, double-homojunction, WZ and KS CZTS nanocrystals. Explicatively, the WZ and KS CZTS nanocrystals with the same composition have been synthesized for reference (Supplementary Figs. 22–25 and Supplementary Table 2). Both of their spectra show a strong absorption in visible region, and the absorption regions of polytypic nanocrystals is between WZ and KS nanocrystals. The insert in Fig. 3a shows the linear extrapolation of $(\alpha h\upsilon)^2$ versus photon energy $(h\upsilon)$ spectra, which are used to evaluate the bandgaps of the synthesized nanocrystals. The bandgaps of double-homojunction and single-homojunction polytypic CZTS nanocrystals are 1.47 and 1.51 eV, which is between the bandgap of KS (1.44 eV) and WZ (1.55 eV) CZTS nanocrystals (Supplementary Table 2). Therefore, the band gaps of the prepared polytypic nanocrystals are suitable for visible-light absorption.

CZTS have been shown as an excellent sunlight absorber not only for solar energy conversion by photovoltaics but also for photocatalytic hydrogen evolution[8,12,43–45]. The synthesized hydrophobic ligand-capped polytypic CZTS could undergo ligand exchange with MPA before photocatalytic hydrogen evolution (Supplementary Fig. 26). The photocatalytic hydrogen evolution properties of polytypic CZTS nanocrystals were performed in a Pyrex reaction cell connected to a closed gas circulation with vacuum. Figure 3b shows the photocatalytic hydrogen evolution rate of the synthesized polytypic and phase-pure CZTS nanocrystals (Supplementary Fig. 27). The samples of double-homojunction and single-homojunction polytypic CZTS nanocrystals boost to a hydrogen production rate of 381 and 270 µmol h$^{-1}$ g$^{-1}$, respectively. Small hydrogen evolution rates (145 µmol h$^{-1}$ g$^{-1}$ for WZ nanocrystals and 98 µmol h$^{-1}$ g$^{-1}$ for KS nanocrystals) are obtained when phase-pure nanocrystals were used as photocatalysts. Apparent quantum efficiencies on DHP CZTS photocatalysts were tested at diverse light wavelengths in the same reaction solution, exhibiting a similar trend to that of the absorption spectra (Supplementary Fig. 28 and Fig. 3a). The best photocatalytic activity of DHP nanocrystals reveals that more homojunctions in polytypic nanocrystals result in better photocatalytic hydrogen evolution performance. The polytypic nanocrystals achieved superior photocatalytic activities compared to prior co-catalyst free CZTS photocatalysts (Supplementary Table 3). Figure 3c exhibits the durability for hydrogen evolution of polytypic nanocrystals. After six runs, no significant decrease for hydrogen evolution can be observed during the long-time irradiation under visible light. Corresponding TEM images, XRD patterns, and XPS spectra (Supplementary Figs. 29–31) after six runs of photocatalysis show no obvious change in morphology, phases and oxidation states of the SHP and DHP CZTS nanocrystals, which certify the high stability of SHP and DHP CZTS nanocrystals against photocorrosion.

To gain the intrinsic reason for the excellent photocatalytic performance of the polytypic CZTS nanocrystals, DFT calculation was further utilized to investigate the electronic band alignment between WZ and KS phases of CZTS[46]. A homojunction interface without significant lattice mismatch can be formed between the (112) surface of

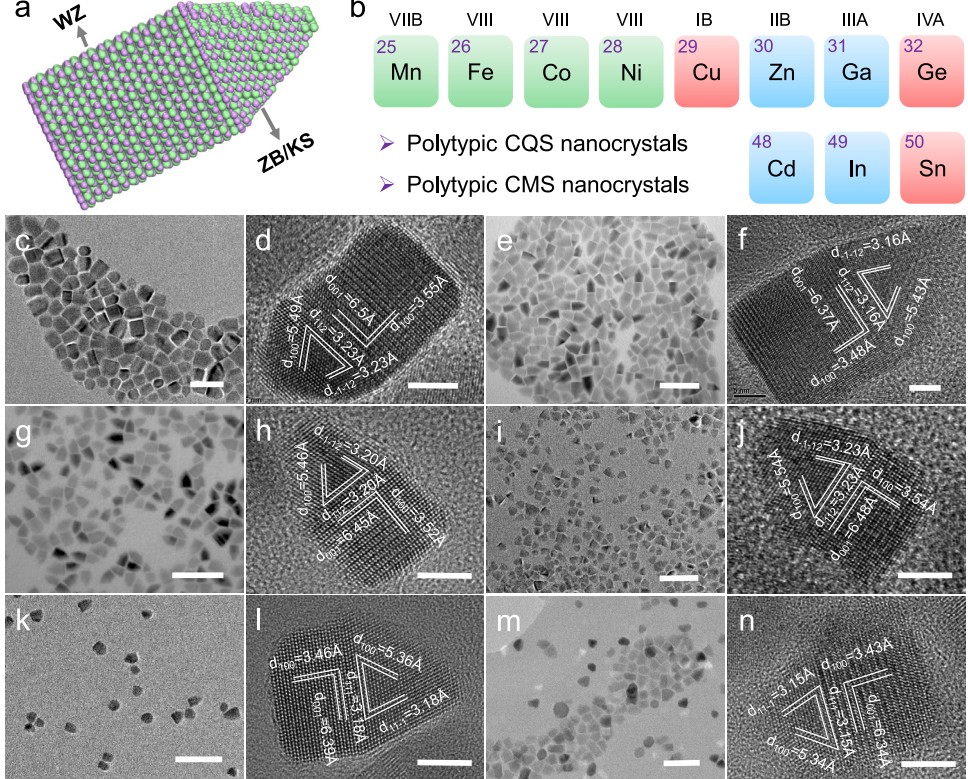

**Fig. 4 | Characterization of the single-homojunction polytypic CQS nanocrystals. a** Schematic graph of a single-homojunction polytypic nanocrystal. **b** Overview of the metal elements involved in the polytypic CQS nanocrystals. TEM (**c**) and HRTEM (**d**) images of polytypic CCdTS nanocrystals. TEM (**e**) and HRTEM (**f**) images of polytypic CCoTS nanocrystals. TEM (**g**) and HRTEM (**h**) images of polytypic CMnTS nanocrystals. TEM (**i**) and HRTEM (**j**) images of polytypic CFeTS nanocrystals. TEM (**k**) and HRTEM (**l**) images of polytypic CInTS nanocrystals. TEM (**m**) and HRTEM (**n**) images of polytypic CGaTS nanocrystals. Scale bars are 50 nm for **c**, **e**, **g**, **i**, **k**, and **m**, 5 nm for **d**, **f**, **h**, **j**, **l**, and **n**, respectively.

the KS structure and the (002) surface of the WZ structure after proper rebuilding the crystal of WZ phase (Supplementary Figs. 1 and 32 and Supplementary Table 4). The calculation details are shown in the experimental method[47–50]. The offset of band gaps disclosed by the Density of States (DOS) of separated bulk KS and WZ phases (Fig. 3d) suggests the possibility of type II band alignment as KS and WZ form a mixture properly. We expect that, when KS and WZ phases form a homojunction along $[112]_{KS}/[002]_{WZ}$ direction, as observed in experiment, a type II band alignment (Fig. 3e) in which both valence band maximum and conduction band minimum of KS(112) are lower than their counterparts of WZ(001) can be made. To verify this idea, we have built a homojunction consisting of KS(112) and WZ(001) phases of CZTS (Supplementary Figs. 32 and 33). The computed band gap (1.60 eV) of the homojunction (Supplementary Fig. 34) is well between that of KS (1.56 eV) and WZ (1.66 eV). Therefore, photogenerated electrons and holes will be accumulated in KS and WZ, respectively, realizing charge separation across the homojunction, as demonstrated by the differential charge profile of the interface (Fig. 3f). Well charge separation of the CZTS homojunction will facilitates its photocatalytic activity. Further, the synergetic enhancement of two homojunctions in double-homojunction polytypic CZTS nanocrystals leads to better photocatalytic performance than that of single-homojunction CZTS nanocrystals.

## Synthesis and photocatalytic applications of other polytypic CQS nanocrystals

Ut supra, the homojunction in polytypic CZTS is benefit for photocatalytic hydrogen evolution. So, designing a photocatalyst with polytypic nanostructure is an effectively way to enhance its performance. It is necessary to promote the synthetic method to prepare a library of polytypic CQS nanocrystals. Fortunately, our synthesis method based on DDT and OLA is widely applicable. We illustrate the generality of such synthetic method by showcasing another six single-homojunction polytypic CQS nanocrystals (Fig. 4a, b). For example, when cadmium acetate dihydrate (Cd(AC)$_2$·2H$_2$O) is used to instead of zinc acetate dihydrate (Zn(AC)$_2$·2H$_2$O) in a solution containing 1 ml of DDT and 8 ml of OLA, single-homojunction polytypic CCdTS nanocrystals were synthesized (Fig. 4c, d and Supplementary Table 5). Similarly, when cobalt (II) acetate tetrahydrate (Co(AC)$_2$·4H$_2$O), manganous acetate (Mn(AC)$_2$), ferrous acetate (Fe(AC)$_2$), indium (III) acetylacetonate (In(acac)$_3$) and gallium (III) acetylacetonate (Ga(acac)$_3$) were used to replace of Zn(AC)$_2$·2H$_2$O, single-homojunction polytypic CCoTS, CMnTS, CFeTS, CInTS and CGaTS nanocrystals would also be synthesized, respectively (Fig. 4e–n and Supplementary Table 5). The HRTEM images in Fig. 4 show that all the synthesized polytypic nanocrystals are formed from a rectangle WZ part and a KS/ZB cusp separated with a homojunction. Besides, the PXRD results directly proved that the obtained polytypic nanocrystals consisted of two different crystal structures—WZ and KS/ZB structures (Supplementary Fig. 35 and Supplementary Table 6). The EDS element mappings and EDS spectra (Supplementary Figs. 36 and 37) show the existence and homogenous distribution of the contained elements in the polytypic nanocrystals, revealing the homojunction is not formed from the element uneven distribution.

Most importantly, this colloidal method is an universal approach which has been successfully used to synthesize more complex copper-based multinary sulfide polytypic nanocrystals, such as single-homojunction polytypic Cu$_2$(Zn$_x$Cd$_y$)SnS$_4$, Cu$_2$(Zn$_x$Co$_y$)SnS$_4$ and Cu$_2$(Zn$_x$Co$_y$Cd$_z$)SnS$_4$ nanocrystals (Supplementary Figs. 38–40 and Supplementary Table 7). The obtained polytypic Cu-based multinary

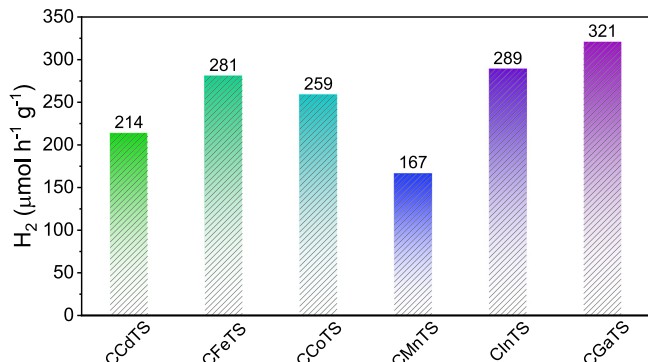

**Fig. 5 | Photocatalytic performances of polytypic CQS nanocrystals.**
Comparison of hydrogen evolution activities of polytypic CQS nanocrystals.

sulfide nanocrystals (CMS) have the same morphology and structure to the polytypic CQS nanocrystals (Supplementary Fig. 38). Besides, the results of X-ray photoelectron spectroscopy (XPS) analysis (Supplementary Fig. 41) indicate that the phase and composition cannot influence the element oxidation state of the obtained nanocrystals.

Among the family of copper-based chalcogenide semiconductors, expect for CZTS, $Cu_2MSnS_4$ (M = Cd, Co, Mn, Fe) and $Cu_3MSnS_5$ (M = In, Ga) are also photo-sensitive with suitable band gaps which makes these proper materials in energy-harvesting applications[51–53]. Since the polytypic homojunction existed in CZTS nanocrystals leads to enhanced photocatalytic performance, we have also investigated the photocatalytic performance of the obtained single-homojunction polytypic $Cu_2MSnS_4$ and $Cu_3MSnS_5$ nanocrystals which have suitable bandgaps for solar harvesting (Supplementary Fig. 42). As shown in Fig. 5, the polytypic nanocrystals exhibited good photocatalytic properties, and the photocatalytic hydrogen evolution rate of polytypic CCdTS, CFeTS, CCoTS, CMnTS, CInTS and CGaTS nanocrystals are 214, 281, 259, 167, 289 and 321 $\mu mol\,h^{-1}\,g^{-1}$, respectively. After 8 h of visible-light irradiation, the amounts of hydrogen produced by polytypic CCdTS, CFeTS, CCoTS, CMnTS, CInTS and CGaTS nanocrystals are 1669, 2198, 2063, 1313, 2288, and 2546 $\mu mol\,g^{-1}$, respectively (Supplementary Fig. 43). No significant decrease for hydrogen evolution rate is observed during the 8 h irradiation under visible light, revealing the good photocatalytic stability. In addition, the other phase-pure CQS nanocrystals with the same composition have been synthesized for references (Supplementary Figs. 44 and 45 and Supplementary Tables 8 and 9). All the other phase pure CQS nanocrystals exhibit lower photocatalytic hydrogen production rates than that of the related polytypic CQS nanocrystals (Supplementary Fig. 46), revealing that polytypic homojunction can enhance the photocatalytic hydrogen evolution performances of semiconductors.

## Discussion

In conclusion, we develop a general colloidal method to construct a library of polytypic copper-based multinary sulfide nanocrystals, which offers exactly controlling over conventional approaches for the synthesis of polytypic nanocrystals. The type-II homojunction formed in polytypic CZTS nanocrystals enables highly efficient solar-to-hydrogen conversion, wherein the photocatalytic activity scales with the homojunction number. Furthermore, the photocatalytic performances of other synthesized polytypic CQS nanocrystals are examined, which show good photocatalytic hydrogen evolution properties, and thus demonstrating that polytypic homojunction can enhance the photocatalytic hydrogen evolution performances of semiconductors. This family of homojunction-based polytypic nanostructures offers a valid and feasible strategy to further enhance and optimize the photocatalytic performances of noble metal free earth-abundance photocatalysts.

## Methods

### Synthesis of polytypic CZTS nanocrystals

In a typical synthesis of single-homojunction polytypic CZTS nanocrystals, CuCl (0.28 mmol), $Zn(AC)_2\cdot 2H_2O$ (0.21 mmol) and $SnCl_2\cdot 2H_2O$ (0.14 mmol) were dissolved in a mix solution with 10 ml of OLA and 1.5 ml of 1-DDT in a 25 ml three-neck flask in air and then heated up to 280 °C at a heating rate of 10 °C/min and kept at 280 °C for 60 min. Then, the flask was removed from the heating mantle and naturally cooled down. The reaction solution was centrifuged at 3552 × g for 5 min and the single-homojunction polytypic CZTS nanocrystals were obtained through discarding the upper clear solution. Then hexane was added to disperse the nanocrystals. To collect the nanocrystals again, ethanol was added into the dispersion and the formed slurry were centrifuged again at 3552 g for 5 min. The nanocrystals were washed through repeating the above dispersing and depositing process for two times. The double-homojunction polytypic CZTS nanocrystals were synthesized by using 0.5 ml of 1-DDT in the same reaction conditions. The detail amounts of the precursors were listed in Supplementary Table 2. The CZTS nanocrystals with different Zn content were synthesized with the increasing amount of $Zn(AC)_2\cdot 2H_2O$ (The detail amounts of the precursors were listed in Supplementary Table 1).

### Synthesis of other polytypic CQS nanocrystals

The synthesis method is the same as that for CZTS polytypic nanocrystals with the replacement of $Zn(AC)_2\cdot 2H_2O$ with $Cd(AC)_2\cdot 2H_2O$, $Co(AC)_2\cdot 4H_2O$, $Mn(AC)_2$, $Fe(AC)_2$, $In(acac)_3$ and $Ga(acac)_3$. The detail amounts of the precursors were listed in Supplementary Table 5.

### Synthesis of copper-based multinary polytypic nanocrystals

The synthesis method is the same as that for CZTS polytypic nanocrystals expect for the added precursors. The detail amounts of the precures were listed in Supplementary Table 7.

### Characterization

Nanocrystals dispersed in hexane were dropped on Mo grid for TEM, HRTEM and HAADF investigation, which were performed on JEOL-2010F and JEM-ARM200F with an acceleration voltage of 200 KV. EDS was carried out on Inca Oxford equipped on JEOL-2010F. The specimen prepared by drop-casting on a Si substrate was characterized by PXRD, using a Philips X'Pert PRO SUPER X-ray diffractometer equipped with graphite monochromatized Cu Kλ radiation ($\lambda = 1.54056$ Å). The operation voltage and current were kept at 40 kV and 400 mA, respectively. The simulate WZ powder XRD pattern were obtained from Diamond 3.2. Raman spectra were recorded with a Renishaw System 2000 spectrometer using the 514 nm line of semiconductor lasers for excitation. UV-Vis-NIR spectroscopy of the polytypic nanocrystals dispersed in tetrachloroethylene were measured at room temperature using a DUV−3700 UV-vis-NIR spectrometer (Shimadzu). XPS was performed on an ESCA Lab MKII XPS using Mg Ka radiation exciting source.

### Ligand exchange

The synthesized nanocrystals (30 mg) with hydrophobic ligands were dissolved in 5 ml chloroform (solution A). In total, 0.5 g KOH was added to 15 ml methanol with 0.5 ml MPA (solution B). Then, the solution A was swiftly added to the solution B. The mixing solution was stirred at room temperature for 5 h. The nanocrystals were collected by centrifugation and washed with water and methanol for twice. The final product was dispersed in water and stored in glovebox.

### Photocatalytic hydrogen evolution test

The photocatalytic hydrogen evolution experiments were performed in a Pyrex reaction cell connected to a closed gas circulation with vacuum. Typically, 30 mg of obtained photocatalyst powder was

suspended in 100 ml of aqueous solution containing 0.25 M $Na_2SO_3$ and 0.35 M $Na_2S$ as sacrificial agents, and subsequently sonicated for 30 min. The reaction solution was evacuated several times to remove air completely prior to irradiation under a 300 W Xe lamp equipped with a 420 nm cut-off filter. The temperature of the reactant solution was maintained at room temperature by a flow of cooling water during the reaction. The amount of hydrogen produced from the photocatalytic reaction was determined using a gas chromatograph (Agilent 7890 A).

## DFT calculation

All spin-polarized DFT calculations have been performed with the Vienna Ab initio Simulation Package. The WZ crystallographic unit cell was a hexagonal structure while the KS was tetragonal. The homojunction of KS(112)-WZ(001) has been built using the following steps. We first rebuild a tetragonal supercell of WZ phase from its hexagonal supercell to make sure both sections of KS(112) and WZ(001) are rectangles with very close side lengths (Supplementary Table 4). Since the lattice mismatch is small (less than 1%), we then place KS(112) and WZ(001) in a proper bonding distance to form the homojunction. The supercell of KS-WZ interface (Supplementary Fig. 33) consists of 4 layers of KS (112) and 3 layers of WZ (001). The Perdew–Burke–Ernzerhofer functional in combination with PAW pseudopotentials and a 500 eV plane-wave kinetic energy cutoff is used for geometry optimization of bulk phases of KS and WZ and their homojunction. A $7 \times 7 \times 3$ Gamma k-point sampling was adopted to calculate the integral in the first Brillouin zone. the convergence criteria for total energy and force were $10^{-5}$ eV and 0.02 eV/Å, respectively. As the total magnetic moment is found to be 0, we then turn off the spin-polarization in the rest of single point calculations. For the calculation of band gaps of all systems, we employ the range-separated hybrid functional, the HSE06 (Heyd-Scuseria-Ernzerhof) functional.

## Reporting summary

Further information on research design is available in the Nature Research Reporting Summary linked to this article.

## Data availability

The data that support the findings of this study are available on request from the corresponding author (S.H.Y.). Source data are provided with this paper.

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

## Acknowledgements

This work was supported by the National Key Research and Develop-ment Program of China (Grants 2021YFA0715700, 2018YFE0202201 and 2018YFA0208702), the National Natural Science Foundation of China (Grants 51732011, U1932213, 21771170, 22101271), the University Synergy Innovation Program of Anhui Province (Grant GXXT-2019-028), Science and Technology Major Project of Anhui Province (201903a05020003). L.W. acknowledges the funding support from the China Postdoctoral Science Foundation (2017M622016 and 2017LH006). The Super-computing Center of University of Science and Technology of China is acknowledged for numerical calculations.

## Author contributions

L.W., T.T.Z., F.J.F. and S.H.Y. conceived the idea. S.H.Y. supervised the project. L.W. carried out the experiments, analyzed the results, and wrote the paper. T.T.Z. and H.H.L. revised the paper. Y.L. helped with the photocatalysis experiments. Q.W. and G.Z.Z. performed the DFT calcu-lation, analyze the data and write the computational portion. All authors discussed the results and assisted during manuscript preparation.

## Competing interests

The authors declare no competing interests.
