## [Peer Review File · Nature Communications]

A library of polytypic copper-based quaternary sulfide nanocrystals enables efficient solar-to-hydrogen conversionREVIEWER COMMENTS

Reviewer #1 (Remarks to the Author):

In this work, the authors reported a general colloidal method to construct polytypic copper-based quaternary sulfide nanocrystals, forming homojunctions between kieserite and wurtzite phases which are beneficial for photocatalytic H₂ evolution. By changing the amount of DDT, CZTS of bullet-shaped single-homojunction (SHP) and rugby-shaped double-homojunction (DHP) can be obtained respectively. Thanks to the type-II bandgap alignment of polytypic homojunction, polytypic CZTS shows better photocatalytic activities than that of phase-pure CZTS, and DHP is superior to SHP. The colloidal method seems matured and my doubts are as follows:

1. In synthesizing polytypic CZTS nanocrystals, Cu(I) and Sn(II) were used as the Cu and Sn source, but according to XPS in Supplementary Figure 28, Cu and Sn ended up as Cu(I) and Sn(IV), showing that Sn was oxidized but Cu was not. Theoretically, the redox potentials of Sn⁴⁺/Sn²⁺ and Cu²⁺/Cu⁺ are very similar, at +0.151 V and +0.153 V, respectively, according to CRC Handbooks of Physics and Chemistry. Why is Sn oxidized during nanocrystal synthesis but Cu not? An additional question about the redox state is that if Cu(II) precursors are used, can well-defined CZTS nanocrystals be obtained?
2. Using 1.5 mL DDT afforded SHP CZTS while DHP CZTS was formed with 0.5 mL DDT. Did the authors attempted to figure out the “turning point”, or the switch of DHP/SHP formation at different DDT dosages? Can multiple homojunctions (3 or more) even be formed under certain synthetic conditions tested so far?
3. When the dosage of 1-DDT was 1 mL, DHP and SHP with uniform size were formed. What is the solar-to-hydrogen performance of this mixed sample?
4. Whether the morphology of the nanocrystal will influence the catalytic performance? Why did the morphology of double-homojunction and single-homojunction nanocrystal are completely different?
5. The synthesized CZTS can undergo ligand exchange with MPA before photocatalytic H₂ evolution. How is the colloidal dispersity or stability of CZTS nanoparticles after MPA ligand exchange and during photocatalysis? I noted that MPA-capped CZTS has to “store in glovebox” while the storage of hydrophobic ligand-capped CZTS has not been specially mentioned. Why does ligand-exchanged CZTS nanocrystals tend to have reduced air-stability?
6. In photocatalytic tests, 0.25 M Na₂SO₃ and 0.35 M Na₂S was used as the sacrificial agent. Why should Na₂SO₃ and Na₂S be used both? Why this concentration and molar ratio?
7. Figure 3a shows that the absorption of DHP in the 1000-1500 nm band was greater than 0. Is DHP absorbed in this region?
8. To demonstrate the concept that polytypic homojunction can enhance the photocatalytic hydrogen evolution performances of semiconductors, solar-to-hydrogen performances of other phase-pure CQS nanocrystals are recommended.

9. The XPS signals of Fe 2p and Mn 2p are too weak (Supplementary Figure 28g-h), and clear XPS spectra are needed.

10. The quantum yield was not provided in the manuscript.

Reviewer #2 (Remarks to the Author):

Wu et. al. report the synthesis and photocatalytic activities of polytypic copper-based multinary sulfide nanocrystals. They argue that the zinc blende and wurtzite twinning leads to a type-II band structure that enables efficient solar-to-hydrogen conversion. I think the general method of synthesis of polytypic copper-based quaternary sulfide nanocrystals with tunable junction structures is of broad interest to the development of high-efficient photocatalysts. The obtained materials have been well characterized in terms of both the structure and physical properties by many instruments and calculation method. I support the publication of the paper on Nature Communications after the following points addressed.

1. The authors provide a general colloidal method to construct a new library of polytypic copper-based quaternary sulfide nanocrystals. Did the authors try to estimate the cost for such materials? Is it possible for large-scale production at low cost? Moreover, there are different ligands were used during the synthesis. A detailed discussion about the formation mechanism of the homojunction would benefit the community tremendously.

2. Another important issue is photocatalytic property of the nanocrystals. For solar hydrogen production from water, there are challenges in obtaining high energy conversion efficiency. The copper-based quaternary sulfides work in the presence of sacrificial reagents such as Na₂S/Na₂SO₃ for hydrogen production. The authors made tunable photocatalytic activity by homojunction design with different junction numbers. This is interesting. However, it is also important to provide the quantum efficiency by detailed experiments and calculation and compare it with reported results. This would make the paper become more interesting.

3. A question raised for discussion. I noted the syntheses were conducted in oil-phase reaction, which indicates that the catalyst surface will be covered oleophilically. How did the authors make sure the catalyst dispersed good in water and conduct the photocatalytic reaction? The authors should also double check the stability of the catalyst, as sulfides are general unstable for photocatalytic hydrogen production.

Manuscript ID: NCOMMS-21-47900

“A library of polytypic copper-based quaternary sulfide nanocrystals enables efficient solar-to-hydrogen conversion”

We have carefully considered all the concerns of the two independent reviewers, and have made suitable revision accordingly. For clearness reason, the answers were marked with RED color and started with “**” and the revision parts in the revised manuscript were noted in RED color.

Reviewer #1 (Remarks to the Author):

In this work, the authors reported a general colloidal method to construct polytypic copper-based quaternary sulfide nanocrystals, forming homojunctions between kieserite and wurtzite phases which are beneficial for photocatalytic H₂ evolution. By changing the amount of DDT, CZTS of bullet-shaped single-homojunction (SHP) and rugby-shaped double-homojunction (DHP) can be obtained respectively. Thanks to the type-II bandgap alignment of polytypic homojunction, polytypic CZTS shows better photocatalytic activities than that of phase-pure CZTS, and DHP is superior to SHP.

****We thank the referee for the comments that encourage us to improve the quality of manuscript. We have now supplemented more experimental data as suggested and carefully revised the manuscript, and sincerely hope that our revisions can address the referee’s concerns.**

The colloidal method seems matured and my doubts are as follows:

1. In synthesizing polytypic CZTS nanocrystals, Cu(I) and Sn(II) were used as the Cu and Sn source, but according to XPS in Supplementary Figure 28, Cu and Sn ended up as Cu(I) and Sn(IV), showing that Sn was oxidized but Cu was not. Theoretically, the redox potentials of Sn⁴⁺/Sn²⁺ and Cu²⁺/Cu⁺ are very similar, at +0.151 V and +0.153 V, respectively, according to CRC Handbooks of Physics and Chemistry. Why is Sn oxidized during nanocrystal synthesis but Cu not? An additional question about the redox state is that if Cu(II) precursors are used, can well-defined CZTS nanocrystals be obtained?

****We thank the referee for these helpful comments. According to the previous reports (*J. Am. Chem. Soc.* **2009**, *131*, 12054–12055; *Langmuir* **2017**, *33* (24), 6151-6158.), the Sn(II) is easily oxidized to Sn(IV) in the OLA/DDT solvent. Besides, The Cu(II) was reduced to Cu(I) when Cu(II) source was used for synthesizing CZTS nanocrystals (*J. Am. Chem. Soc.* **2010**, *132* (37), 12778-12779; *J. Am. Chem. Soc.* **2011**, *133* (40), 15910-3). Based on this, although the redox potentials of Sn⁴⁺/Sn²⁺ and Cu²⁺/Cu⁺ are very similar the Cu and Sn ended up as Cu(I) and Sn(IV) in the obtained polytypic nanocrystals.**

As suggested by the Reviewer, we have used Cu(II) precursors, such as $\text{CuCl}_2 \cdot 2\text{H}_2\text{O}$, $\text{Cu}(\text{acac})_2$, and $\text{Cu}(\text{AC})_2$, to synthesize polytypic CZTS nanocrystals in the same reaction conditions. When $\text{CuCl}_2 \cdot 2\text{H}_2\text{O}$ was used for synthesis of SHP and DHP CZTS nanocrystals, the irregular shape CZTS nanocrystals were obtained (Figure R1a-d, Figure R2). CZTS nanorods would be synthesized by using $\text{Cu}(\text{acac})_2$ as Cu precursor (Figure R1e-h, Figure R2). When $\text{Cu}(\text{AC})_2$ was used as Cu precursor, bullet-shape polytypic CZTS nanocrystals were obtained with 0.5 mL of 1-DDT and CZTS nanoplates were produced with 1.5 mL of 1-DDT (Figure R1i-l, Figure R2). In conclusion, well-defined polytypic CZTS nanocrystals would be synthesized using specific Cu(II) precursors.

Figure R1 (new Supplementary Figure 13). TEM and magnified TEM images of polytypic CZTS nanocrystals synthesized with different Cu sources. a-b, CuCl_2 , 0.5 mL of 1-DDT. c-d, CuCl_2 , 1.5 mL of 1-DDT. e-f, $\text{Cu}(\text{acac})_2$, 0.5 mL of 1-DDT. g-h, $\text{Cu}(\text{acac})_2$, 1.5 mL of 1-DDT. i-j, $\text{Cu}(\text{AC})_2$, 0.5 mL of 1-DDT. k-l, $\text{Cu}(\text{AC})_2$, 1.5 mL

of 1-DDT. Scale bars are 200 nm for **a, c, e, g, i, and k**, 50 nm for **b, d, f, h, j and l**, respectively.

Figure R2 (new Supplementary Figure 14). XRD patterns of polytypic CZTS nanocrystals synthesized with different Cu sources.

2. Using 1.5 mL DDT afforded SHP CZTS while DHP CZTS was formed with 0.5 mL DDT. Did the authors attempted to figure out the “turning point”, or the switch of DHP/SHP formation at different DDT dosages? Can multiple homojunctions (3 or more) even be formed under certain synthetic conditions tested so far?

****Thanks to the valuable suggestions from the Reviewer. We further studied the reaction products of CZTS nanocrystals obtained with different amounts of 1-DDT. When 1.5 mL of 1-DDT was used as S precursor, the obtained nanocrystals are SHP polytypic CZTS nanocrystals (Figure 1a-b) without rugby-shaped DHP nanocrystals. The obtained nanocrystals are almost DHP CZTS nanocrystals with some bullet-shaped SHP nanocrystals when 1.5 mL of 1-DDT was used as S precursor. The obtained CZTS**

nanocrystals have an irregular morphology with 0.2 mL of 1-DDT (Supplementary Fig. 5a-b). Both bullet-shaped and rugby-shaped nanocrystals have been obtained with 1.0 mL of 1-DDT (Supplementary Fig. 5a-b). Therefore, it is very difficult to figure out the turning point of the switch of DHP/SHP formation.

The SHP and DHP CZTS nanocrystals are synthesized by epitaxial growth of KS structure on WZ structure. Because only two {0001} facets of WZ structure match well with the {111} facets of KS structure and can be acted as growth substrates, the obtained polytypic nanocrystals would only have one or two homojunctions. Multiple homojunctions (3 or more) could not be formed under this synthetic conditions. Interestingly, tetrapod Cu_2SnSe_3 nanocrystals with four homojunctions have been successfully synthesized under certain synthetic conditions (*J. Am. Chem. Soc.* **2013**, 135 (21), 7835-8; *J. Am. Chem. Soc.* **2014**, 136 (22), 7954-60.).

3. When the dosage of 1-DDT was 1 mL, DHP and SHP with uniform size were formed. What is the solar-to-hydrogen performance of this mixed sample?

**Thank you for your helpful suggestion. We have tested the photocatalytic hydrogen production performance of the mixed sample. As shown in Figure R3, the photocatalytic hydrogen production rate of the mixed sample is higher than the SHP nanocrystals but lower than the DHP nanocrystals, which is consistent with our conclusion that DHP benefits the charge separation and solar-to-hydrogen production.

Figure R3 (new Supplementary Figure 27). Photocatalytic H₂ activities of polytypic CZTS nanocrystals, mixed sample and phase-pure CZTS nanocrystals.

4. Whether the morphology of the nanocrystal will influence the catalytic performance? Why did the morphology of double-homojunction and single-homojunction nanocrystal are completely different?

Thanks for your valuable suggestions. The morphology of the semiconductor nanocrystals has influences on the catalytic performances (*Nat. Commun.* **2020, 11 (1), 5194; *Chem. Rev.* **2010**, 110 (11), 6503-6570.). In terms of the polytypic nanocrystals presented here, there is no significant morphology difference between DHP and SHP CZTS nanocrystals, which is thus believed to show negligible influence on their catalytic performances.

All the DHP and SHP nanocrystals are obtained through epitaxial growth of KS phase on the (0001) facets of WZ structures. The SHP CZTS nanocrystals are obtained through epitaxial growth of KS phase on the $-(0001)$ facet, which have only one homojunction and exhibit a bullet shape. While the DHP CZTS nanocrystals are

synthesized by epitaxial growth of KS on both two {0001} facets, which have two homojunctions and thus exhibit a rugby shape. As a result, morphologies of DHP and SHP nanocrystal are different to some extent, but both are quasi-zero dimensional nanocrystals and should show little morphological influence on their catalytic performances.

5. The synthesized CZTS can undergo ligand exchange with MPA before photocatalytic H₂ evolution. How is the colloidal dispersity or stability of CZTS nanoparticles after MPA ligand exchange and during photocatalysis? I noted that MPA-capped CZTS has to “store in glovebox” while the storage of hydrophobic ligand-capped CZTS has not been specially mentioned. Why does ligand-exchanged CZTS nanocrystals tend to have reduced air-stability?

We thank the referee for these helpful comments. We have characterized the stability of CZTS nanoparticles after MPA ligand exchange. As shown in Figure R4a-b, the hydrophobic ligand-capped CZTS dispersed well in hexane, and the MPA-capped CZTS dispersed well in water (*ACS Appl. Mater. Interfaces* **2015, 7 (32), 17623-9.). The TEM images of the MPA-capped SHP and DHP CZTS nanocrystals (Figure R3c-f) show that the nanocrystals dispersed well in water without aggregation.

The hydrophobic ligand-capped CZTS dispersed in hexane/toluene are stored in sealed vial.

The MPA ligands prefers to react with the oxygen in the air, resulting to reduce the dispersity of the CZTS nanocrystals (*Chem. Mater.* **2021**, 33 (3), 892-901; *J. Am. Chem. Soc.* **2014**, 136 (26), 9236-9), storing the MPA-capped CZTS nanocrystals in glovebox is the best for keeping the stability of CZTS nanocrystals.

Figure R4 (new Supplementary Figure 26). Characterization of the SHP and DHP CZTS nanocrystals after ligand exchange. a-b, Photographs illustrating the phase transfer of SHP and DHP CZTS nanocrystals induced by ligand exchange process. **c-d,** TEM and enlarged TEM images of DHP CZTS nanocrystals dispersed in methanol. **e-f,** TEM and enlarged TEM images of SHP CZTS nanocrystals dispersed in methanol. Scale bars are 200 nm for **c** and **e**, 50 nm for **d** and **f**, respectively.

6. In photocatalytic tests, 0.25 M Na_2SO_3 and 0.35 M Na_2S was used as the sacrificial agent. Why should Na_2SO_3 and Na_2S be used both? Why this concentration and molar ratio?

****We thank the referee for these helpful comments. A mixed solution containing Na_2SO_3 and Na_2S as electron donors were employed for reactant solutions. The photocatalytic activity was efficiently improved through using the mixed solution containing Na_2SO_3 and Na_2S (*J. Am. Chem. Soc.* **2004**, 126 (41), 13406-13413, *J. Phys. Chem. B* **2005**, 109 (15), 7323-7329.). Under this condition, the photocatalytic reaction is supposed to proceed along the following processes:**

H₂O is reduced to H₂ by the photogenerated electrons accompanied by oxidation of sacrificial reagents. Meanwhile, the photogenerated holes oxidize SO₃²⁻ and S²⁻ to SO₄²⁻ and S₂²⁻ directly. The S₂²⁻ ions were efficiently suppressed by mixing with SO₃²⁻ ions. The presence of excess S²⁻ ions in the reaction solution also stabilizes the photocatalyst surface because the formation of sulfur defects could be suppressed. In conclusion, a mix solution of Na₂SO₃ and Na₂S is beneficial for photocatalytic hydrogen production.

The mix solution of 0.25 M Na₂SO₃ and 0.35 M Na₂S is the generally used for photocatalytic hydrogen production test (*Adv. Mater.* **2014**, 26 (21), 3496-500; *Chem. Eur. J.* **2015**, 21 (26), 9514-9; *Nat. Commun.* **2020**, 11 (1), 5194; *Appl. Catal. B-Environ.* **2021**, 298, 120570; *J. Phys. Chem. C* **2020**, 124 (11), 5934-5943). Therefore, we used this concentration and molar ratio of Na₂SO₃ and Na₂S for characterizing the photocatalytic performances of polytypic CZTS nanocrystals.

7. Figure 3a shows that the absorption of DHP in the 1000-1500 nm band was greater than 0. Is DHP absorbed in this region?

**Thanks for your valuable suggestions. The absorption of DHP in the 1000-1500 nm band larger than 0 was due to light scattering caused by the relatively poor dispersity of DHP nanocrystals. As shown in Figure 3a, the almost horizontal absorption curve in the region of 1000-1500 nm illustrated that the DHP nanocrystals were not absorbed in this region.

8. To demonstrate the concept that polytypic homojunction can enhance the photocatalytic hydrogen evolution performances of semiconductors, solar-to-hydrogen performances of other phase-pure CQS nanocrystals are recommended.

**Thanks for your helpful suggestion. The other phase-pure CQS nanocrystals have been synthesized by colloidal method for references (Table R1-R2). The HRTEM images (Figure R5) and the PXRD results (Figure R6) show that all the synthesized CQS nanocrystals are phase-pure nanocrystals.

Furthermore, the solar-to-hydrogen performances of the synthesized phase-pure CQS nanocrystals are characterized for comparison with the related polytypic CQS nanocrystals. As shown in Figure R7a-b, WZ and KS CCdTS nanocrystals exhibit the

photocatalytic hydrogen production rates of 151 and 89 $\mu\text{mol h}^{-1} \text{g}^{-1}$, which are lower than that of polytypic CCdTS nanocrystals. In addition, all the other phase pure CQS nanocrystals exhibit lower photocatalytic hydrogen production rate than that of the related polytypic CQS nanocrystals. These results illustrate that polytypic homojunction can enhance the photocatalytic hydrogen evolution performances of semiconductors.

Table R1 (new Supplementary Table 7). Amounts of precursors, 1-DDT and OLA used for synthesizing the WZ CQS nanocrystals

Sample	M (mmol)	Cu(acac) ₂ /Sn(acac) ₂ Cl ₂ (mmol)	OLA/1-DDT/t-DDT (mL)	Reaction Temperature
WZ CCdTS	Cd(acac) ₂ (0.5)	1/0.5	12/0.37/2.63	220 °C
WZ CCoTS	Co(acac) ₃ (0.5)	1/0.5	12/0.37/2.63	210 °C
WZ CFeTS	Fe(acac) ₃ (0.4)	1/0.5	12/0.37/2.63	210 °C
WZ CMnTS	Mn(acac) ₃ (0.4)	1/0.5	12/0.37/2.63	210 °C
WZ CInTS	In(acac) ₃ (0.5)	1.5/0.5	12/0.25/2.75	220 °C
WZ CGaTS	Ga(acac) ₃ (0.5)	1.5/0.5	12/0.25/2.75	220 °C

Table R2 (new Supplementary Table 8). Amounts of precursors, and OLA used for synthesizing the KS/ZB CQS nanocrystals

Sample	M (mmol)	CuCl/SnCl ₂ (mmol)	OLA (mL)	S (mmol)
KS CCdTS	Cd(AC) ₂ (0.28)	0.56/0.28	15	3
KS CCoTS	Co(AC) ₂ (0.28)	0.56/0.28	15	3
KS CFeTS	Fe(AC) ₂ (0.24)	0.56/0.28	15	3
KS CMnTS	Mn(AC) ₂ (0.24)	0.56/0.28	15	3
KS CInTS	In(AC) ₃ (0.2)	0.6/0.2	15	3
KS CGaTS	Ga(acac) ₃ (0.2)	0.6/0.2	15	3

Figure R5 (new Supplementary Figure 41). TEM and HRTEM images of the synthesized WZ and KS/WZ CQS nanocrystals. a-b, WZ CCdTS nanocrystals. c-d, KS CCdTS nanocrystals. e-f, WZ CCoTS nanocrystals. g-h, KS CCoTS nanocrystals. h-i, WZ CFeTS nanocrystals. j-k, KS CFeTS nanocrystals. l-m, WZ CMnTS nanocrystals. n-o, KS CMnTS nanocrystals. p-q, WZ CInTS nanocrystals. r-s, KS CInTS nanocrystals. t-u, WZ CGaTS nanocrystals. v-w, KS CGaTS nanocrystals. Scale bars are 50 nm for a, c, h, l, n, t, and v, 100 nm for e, g, p, and r, 20 nm for j, 5 nm for b, d, f, h, i, k, m, o, q, s, u, and w, respectively.

Figure R6 (new Supplementary Figure 42). XRD patterns of the synthesized WZ and KS/WZ CQS nanocrystals. a, WZ and KS CCTS. b, WZ and KS CCoTS. c, WZ and KS CFeTS. d, WZ and KS CMnTS. e, WZ and ZB CInTS. f, WZ and ZB CGaTS.

Figure R7 (new Supplementary Figure 43). Photocatalytic hydrogen production properties of the synthesized WZ and KS/WZ CQS nanocrystals. a-b, WZ and KS CCTS. c-d, WZ and KS CCoTS. e-f, WZ and KS CFeTS. g-h, WZ and KS CMnTS. i-j, WZ and ZB CInTS. k-l, WZ and ZB CGaTS.

9. The XPS signals of Fe 2p and Mn 2p are too weak (Supplementary Figure 28g-h), and clear XPS spectra are needed.

****Thanks for your valuable suggestion. The clear XPS signals of Fe 2p and Mn 2p are shown in Figure R8.**

Figure R8 (new Supplementary Figure 38g-h). a-b, XPS spectra of Fe 2P and Mn 2P.

10. The quantum yield was not provided in the manuscript.

**Thank you for your helpful suggestion. The apparent quantum efficiency (AQE) was measured under the identical photocatalytic reactions by using 420 nm, 475 nm, 500 nm, 600 nm, and 650 nm band pass filters, respectively. The trend in apparent quantum efficiency closely followed that of the absorbance measured by ultraviolet–visible spectroscopy (Figure R9, Figure 3a), revealing bandgap-transition-dependent hydrogen evolution behavior.

Figure R9 (new Supplementary Figure 28). Photocatalytic efficiency of DHP CZTS nanocrystals. Apparent quantum efficiency in photocatalytic H_2 production of the DHP CZTS nanocrystals.

Reviewer #2 (Remarks to the Author):

Wu et. al. reports the synthesis and photocatalytic activities of polytypic copper-based multinary sulfide nanocrystals. They argue that the zinc blende and wurtzite twinning leads to a type-II band structure that enables efficient solar-to-hydrogen conversion. I think the general method of synthesis of polytypic copper-based quaternary sulfide nanocrystals with tunable junction structures is of broad interest to the development of high-efficient photocatalysts. The obtained materials have been well characterized in terms of both the structure and physical properties by many instruments and calculation method. I support the publication of the paper on Nature Communications after the following points addressed.

** We thank the Reviewer for the encouragements on the significance and high quality of this work.

1. The authors provide a general colloidal method to construct a new library of polytypic copper-based quaternary sulfide nanocrystals. Did the authors try to estimate the cost for such materials? Is it possible for large-scale production at low cost? Moreover, there are different ligands were used during the synthesis. A detailed discussion about the formation mechanism of the homojunction would benefit the community tremendously.

**Thanks for your valuable suggestion. The cost of polytypic nanocrystals is about 12 USD/g (OLA: 24.8 USD/500mL, 1-DDT: 24.8 USD/250mL, CuCl: 19.8 USD/500g, Zn(AC)₂·2H₂O: 6.6 USD/500g, SnCl₂·2H₂O: 35.8 USD/500g).

As suggested by the Reviewer, we have sought to achieve the large-scale production polytypic CZTS nanocrystals at low cost. Firstly, we increase the concentration of precursors without changing the amount of OLA and 1-DDT for synthesizing SHP CZTS nanocrystals (Table R3). As shown in Figure R10, irregular CZTS nanocrystals would be produced with increasing concentration of precursors by 10 times. The XRD patterns (Figure R11) indicate that impurity peaks have appeared with increasing concentration of precursors by 10 times. Hence, we try to large-scale synthesis of SHP CZTS nanocrystals in a mix solution with 100 mL of OLA and 20 mL of 1-DDT (Table R3). Figure R12a shows the reaction flask with the capacity of 250 mL contains metal cation precursors dissolved in 100 mL of OLA and 20 mL of 1-DDT. The photograph (Figure R12b) shows that the yield of the SHP CZTS nanocrystals after one single reaction is more than 3g. The obtained nanocrystals are nearly monodispersed, and most of the nanoparticles display a bullet shape morphology (Figure R12c-e), illustrating that it is possible for large-scale production polytypic CZTS nanocrystals at low cost.

Furthermore, the formation mechanism of SHP CZTS nanocrystals has been investigated in detail. The time-dependent experiments have been conducted to reveal the growth process of SHP polytypic CZTS nanocrystals. First, the Cu₂S nanocrystals

nucleate at low temperature (Figure R14a-c, Figure 15), followed by diffusion of Zn and Sn ions into Cu_2S nanocrystals to form wurtzite CZTS nanocrystals and nanocylinders (Figure R14d-g, Figure 15), and then KS CZS nucleates on the $-(0001)$ facet of wurtzite CZTS to form a bullet-shaped polytypic nanocrystals with only one homojunctions (Figure R14h-k, Figure 15).

Table R3. Amounts of CuCl , $\text{Zn}(\text{AC})_2 \cdot 2\text{H}_2\text{O}$, $\text{SnCl}_2 \cdot 2\text{H}_2\text{O}$, 1-DDT and OLA used for large-scale synthesis of SHP CZTS nanocrystals.

Sample	Cu (mmol)	Zn (mmol)	Sn (mmol)	OLA (mL)	1-DDT (mL)
G1	1.12	0.84	0.56	10	2
G2	1.68	1.26	0.84	10	2
G3	2.24	1.68	1.12	10	2
G4	2.8	2.1	1.4	10	2
G5	22.4	16.8	11.2	100	20

Figure R10. TEM images of the SHP CZTS nanocrystals synthesized with increased concentration of cation precursors. a-b, G1. c-d, G2. e-f, G3. e-f, G4. Scale bars are 100 nm for a, c, e, and g, 50 nm for b, d, f and h, respectively. The detail amounts of cation precursors were listed in Supplementary Table 4.

Figure R11. TEM images of the SHP CZTS nanocrystals synthesized with increased concentration of cation precursors.

Figure R12 (new Supplementary Figure 21). Characterization of the large-scale synthesized SHP nanocrystals. a, The photograph of the reaction solution color at 180 °C. **b,** Photograph of 3.41 g of SHP nanocrystals powder. **c,** XRD patterns. **d-e,** TEM images. Scale bars are 200 nm for **a** 50 nm for **b**, respectively.

Figure R13 (new Supplementary Figure 3) a, Schematic illustration of the growth mechanisms of SHP nanocrystals. **b-i**, TEM and HRTEM images of the nanocrystals obtained at different stages of the synthesis process of the SHP nanocrystals: **b-c**, The reaction temperature reached 260 °C. **d-e**, The reaction reached 280 °C. **f-g**, The reaction was kept at 280 °C for 5 min. **h-i**, The reaction was kept at 280 °C for 15 min. Scale bars are 200 nm for **b, d, f** and **h**, 50 nm for **c, e, g** and **i**, respectively.

Figure R14 (new Supplementary Figure 4). XRD patterns of the nanocrystals obtained at different stages of the synthesis process of the SHP nanocrystals.

2. Another important issue is photocatalytic property of the nanocrystals. For solar hydrogen production from water, there are challenges in obtaining high energy conversion efficiency. The copper-based quaternary sulfides work in the presence of sacrificial reagents such as Na₂S/Na₂SO₃ for hydrogen production. The authors made tunable photocatalytic activity by homojunction design with different junction numbers. This is interesting. However, it is also important to provide the quantum efficiency by detailed experiments and calculation and compare it with reported results. This would make the paper become more interesting.

****Thanks for your valuable suggestion. The apparent quantum efficiency in photocatalytic H₂ production of the DHP CZTS nanocrystals have been characterized and shown in Figure R9. We have also compared the photocatalytic performances of the polytypic nanocrystals and listed in Supplementary Table 3.**

3. A question raised for discussion. I noted the syntheses were conducted in oil-phase reaction, which indicates that the catalyst surface will be covered oleophilically. How did the authors make sure the catalyst dispersed good in water and conduct the photocatalytic reaction? The authors should also double check the stability of the catalyst, as sulfides are general unstable for photocatalytic hydrogen production.

****Thanks for your valuable suggest. As shown in Figure R3, The CZTS nanocrystals transformed into water after ligand exchange with MPA. The TEM images of the MPA-capped SGP and DHP CZTS nanocrystals (Figure R3c-f) show that the nanocrystals well dispersed in water without aggregation.**

TEM, XRD and XPS were used to characterized the SHP and DHP CZTS nanocrystals after photocatalytic testing. As shown in Figure R15a, d, the SHP and DHP CZTS nanocrystals dispersed well in the aqueous solution of 0.25 M Na_2SO_3 and 0.35 M Na_2S after photocatalytic testing. Corresponding TEM images, XRD patterns, and XPS spectra after 24 h of photocatalysis show no obvious change in morphology, phases and oxidation states of the SHP and DHP CZTS nanocrystals, which certify the high stability of SHP and DHP CZTS nanocrystals against photocorrosion.

Figure R15 (new Supplementary Figure 29) Characterization of the SHP and DHP CZTS nanocrystals after photocatalytic tests. a, Photograph of the DHP CZTS dispersed in water. b-c, TEM and enlarged TEM images of DHP CZTS nanocrystals. c, Photograph of the SHP CZTS dispersed in water. b-c, TEM and enlarged TEM images of SHP CZTS nanocrystals. Scale bars are 200 nm for b and e, 50 nm for c and f, respectively.

Figure R16 (new Supplementary Figure 30) XRD patterns of the SHP and DHP CZTS nanocrystals after photocatalytic tests.

Figure R17 (new Supplementary Figure 31). XPS spectra of the SHP and DHP CZTS nanocrystals after photocatalytic tests. a, Cu_{2p}. b, Zn_{2p}. c, Sn_{3d}. d, S_{2p}.

REVIEWER COMMENTS

Reviewer #1 (Remarks to the Author):

In the revised manuscript, the authors have effectively addressed my questions, and I think the work is publishable in the current form.

Reviewer #2 (Remarks to the Author):

The revised manuscript has well addressed all the comments raised by the reviewers. The current version can be accepted for publication.

Reviewer #3 (Remarks to the Author):

Review of Nature Communications manuscript NCOMMS-21-47900A

I was asked by the editor to review the work's computational portions. The work has undergone one round of review, however, in the previous round of review, the referee with computational experience was unable to provide a report or comment on the computational portions.

The computational part is rather simple: the authors used DFT calculations to estimate band alignment at the interface between KS and WZ phases of CZTS. Band alignment is very difficult to measure experimentally, whereas DFT calculations are well suited to this purpose. However, the calculations presented in the manuscript are very vaguely described and even seem to be wrongly executed (hard to judge due to the vague description):

1) the authors state that they calculated the interface between the (112) surface of KS and (002) of the WZ but do not give any further details. Did the respective lattices match exactly? If not, how was the strain at the interface mitigated? Based on which arguments do the authors believe that this is actual interface geometry in the experimental samples?

2) The results are displayed in the form of density of states (Figure 3e). However, the position of the Fermi energy is not displayed which makes the figure rather useless. The fonts in the figure are so small that they are almost unreadable.

3) The description of calculations in Methods is inadequate. Was the exchange-correlation functional HSE06 or PBE (both are mentioned without specific details)? How was the supercell constructed when WZ had hexagonal unit cell and KS tetragonal one? How many atoms (or atomic layers) were in each part of the supercell?

4) The level of language in computational section is poor making it hard to digest. I would expect very different level of language from a Nature family paper.

Based on these arguments, I cannot recommend the manuscript for publication in the present form.

Manuscript ID: NCOMMS-21-47900A

“A library of polytypic copper-based quaternary sulfide nanocrystals enables efficient solar-to-hydrogen conversion”

We have carefully considered all the concerns of reviewer 3, and have made suitable revision accordingly. For clearness reason, the answers were marked with RED color and started with “**” and the revision parts in the revised manuscript were noted in RED color.

Reviewer #1 (Remarks to the Author):

In the revised manuscript, the authors have effectively addressed my questions, and I think the work is publishable in the current form.

****We thank the reviewer for strong support on the publication of this work.**

Reviewer #2 (Remarks to the Author):

The revised manuscript has well addressed all the comments raised by the reviewers. The current version can be accepted for publication.

****Thanks for the reviewer’s support on the publication of this work.**

Reviewer #3 (Remarks to the Author):

Review of Nature Communications manuscript NCOMMS-21-47900A

I was asked by the editor to review the work's computational portions. The work has undergone one round of review, however, in the previous round of review, the referee with computational experience was unable to provide a report or comment on the computational portions.

The computational part is rather simple: the authors used DFT calculations to estimate band alignment at the interface between KS and WZ phases of CZTS. Band alignment is very difficult to measure experimentally, whereas DFT calculations are well suited to this purpose. However, the calculations presented in the manuscript are very vaguely described and even seem to be wrongly executed (hard to judge due to the vague description):

****We thank the referee for the comment that prompts us to improve the quality of manuscript. We have conducted supplementary calculations as suggested and carefully revised the manuscript, and sincerely hope that our revisions have addressed the referee’s concerns.**

1. The authors state that they calculated the interface between the (112) surface of KS and (002) of the WZ but do not give any further details. Did the respective lattices match exactly? If not, how was the strain at the interface mitigated? Based on which

arguments do the authors believe that this is actual interface geometry in the experimental samples?

****Thanks for the reviewer's helpful comment. We first validate the bulk models of KS and WZ phases adopted in computation by comparing the experimental and references (*Chem. Mater.* **2016**, 28 (3), 720-726; *ACS Nano* **2013**, 7 (2), 1454-1463) and simulated XRD patterns (Figure R1 and Supplementary Figure 24). As shown in Figure 1c, the polytypic CZTS nanocrystal consists of WZ and KS structures. The HRTEM image in Fig. 1h clearly presents two different atom stacking forms separated by one homojunction, such as the A-B-A WZ atom stacking in the rectangle area and the A-B-C KS atom stacking in the cusp. The WZ and KS structures are arranged along $[001]_{\text{WZ}}$ and $[112]_{\text{KS}}$ direction. As a result, KS(112) and WZ(001) form the homojunction that is associated with photocatalytic activity.**

Since KS has a tetragonal lattice while WZ has a hexagonal one, we rebuild a tetragonal supercell of WZ phase from its hexagonal supercell (Figure R2). As a result, both sections of KS(112) and WZ(001) are rectangles with very close side lengths (Table R1), which minimize the lattice mismatch of KS(112)-WZ(001) interface in our computation model. We have elaborate the description of building the KS(112) - WZ(001) interface model in the DFT Calculation section of the revised manuscript in page 17.

Figure R1 (new Supplementary Figure 1). Structure models of CZTS as well as their XRD patterns, respectively. a-b, Wurtzite (WZ). c-d, Kesterite (KS).

Figure R2 (new Supplementary Figure 32). Rebuild the WZ structure along (001) facet, from hexagonal to tetragonal.

Table R1 (new Supplementary Table 4). Lattice constants matching between the KS (112) and WZ (001) facets.

Lattice constant	a (nm)	b (nm)
KS (112)	0.761	1.317
WZ (001)	0.762	1.328

2. The results are displayed in the form of density of states (Figure 3e). However, the position of the Fermi energy is not displayed which makes the figure rather useless. The fonts in the figure are so small that they are almost unreadable.

**Thanks for the reviewer's helpful suggestion. We have modified the figures as suggested, including the addition of CBM, VBM and increasing the size of fonts (Figure R3a). The offset of band gaps disclosed by the Density of States (DOS) of separated bulk KS and WZ phases (Figure R3a) suggests the possibility of type II band alignment as KS and WZ form a mixture properly. We expect that, when KS and WZ phases form a homojunction along $[112]_{\text{KS}}/[002]_{\text{WZ}}$ direction, as observed in experiment, a type II band alignment (Figure R2b) in which both valence band maximum and conduction band minimum of KS(112) are lower than their counterparts of WZ(001) can be made. The computed band gap (1.60 eV) of the homojunction (Figure R4) is well between that of KS (1.56 eV) and WZ (1.66 eV). We also simulate the charge distribution at the interface of the KS-WZ combination. As shown in Figure R3c, photogenerated electrons and holes will be accumulated in KS and WZ, respectively, realizing charge separation across the homojunction.

Figure R3 (new Figure 3e-f). **a**, The calculation Density of States of WZ and KS phases in the polytypic structures. **b**, the bandgap alignments of polytypic nanocrystals. **c**, The simulated charge distributions of the new hybrid combination.

Figure R4 (new Supplementary Figure 34) The simulated DOS of the homojunction of KS(112) and WZ(001) of CZTS.

3. The description of calculations in Methods is inadequate. Was the exchange-correlation functional HSE06 or PBE (both are mentioned without specific details)? How was the supercell constructed when WZ had hexagonal unit cell and KS tetragonal one? How many atoms (or atomic layers) were in each part of the supercell?

****Thanks for the reviewer's suggestion that helps us improve the quality of manuscript. More computation details have been added to the Methods section of main text. We first use PBE functional to optimize the structure and then calculate the density of states (DOS) using HSE06 functional. As for the supercell model of KS-WZ interface, we first rebuild a tetragonal supercell of WZ phase from its hexagonal supercell to make sure both sections of KS(112) and WZ(001) are rectangles with very close side lengths (Figure R1 and Table R1). Since the lattice mismatch is small (less than 1%), we then place KS(112) and WZ(001) in a proper bonding distance to form the homojunction. The supercell of KS-WZ interface consists of 4 layers of KS (112) and 3 layers of WZ (001) (Figure R5).**

Figure R5 (new Supplementary Figure 33). Structure models of the supercell model of KS-WZ interface

4. The level of language in computational section is poor making it hard to digest. I would expect very different level of language from a Nature family paper.

****Thank for the reviewer's critical comment. We have revised the computational section and substantially improve the quality of writing.**

REVIEWERS' COMMENTS

Reviewer #3 (Remarks to the Author):

The authors considered all of my concerns and reviewed the manuscript accordingly. The manuscript is ready for publication.

Manuscript ID: NCOMMS-21-47900B

“A library of polytypic copper-based quaternary sulfide nanocrystals enables efficient solar-to-hydrogen conversion”

Reviewer #3 (Remarks to the Author):

The authors considered all of my concerns and reviewed the manuscript accordingly. The manuscript is ready for publication.

****We thank the reviewer for strong support on the publication of this work.**